# Complementary Gene Therapy after Revascularization with the Saphenous Vein in Diabetic Foot Syndrome

**DOI:** 10.3390/genes14101968

**Published:** 2023-10-20

**Authors:** Diana Kupczyńska, Paweł Lubieniecki, Maciej Antkiewicz, Jan Barć, Katarzyna Frączkowska-Sioma, Tomasz Dawiskiba, Tadeusz Dorobisz, Małgorzata Małodobra-Mazur, Dagmara Baczyńska, Konrad Pańczak, Wojciech Witkiewicz, Dariusz Janczak, Jan Paweł Skóra, Piotr Barć

**Affiliations:** 1Clinical Department of Vascular, General and Transplantation Surgery, Wroclaw Medical University, Borowska Street 213, 50-556 Wroclaw, Poland; dr.diana.kupczynska@gmail.com (D.K.); maciej.antkiewicz@gmail.com (M.A.); 880511kemf@gmail.com (K.F.-S.); tomaszdawiskiba@gmail.com (T.D.); tdorobisz@gmail.com (T.D.); dariusz.janczak@umed.wroc.pl (D.J.); jpskora@gmail.com (J.P.S.); barc.wroclaw@gmail.com (P.B.); 2Clinical Department of Diabetology and Internal Disease, Wroclaw Medical University, Borowska Street 213, 50-556 Wroclaw, Poland; 3Faculty of Medicine, Medical University of Lublin, Aleje Racławickie 1, 20-059 Lublin, Poland; jan8876@gmail.com; 4Department of Forensic Medicine, Division of Molecular Techniques, Wroclaw Medical University, Borowska Street 213, 50-556 Wroclaw, Poland; malgorzata.malodobra-mazur@umed.wroc.pl; 5Department of Molecular and Cellular Biology, Wroclaw Medical University, Borowska 211A, 50-556 Wrocław, Poland; dagmara.baczynska@umed.wroc.pl; 6Lecran Wound Care Center, Trawowa 63a, 54-614 Wrocław, Poland; konrad.panczak@lecran.pl; 7Research and Development Center, Regional Specialized Hospital in Wroclaw, Kamienskiego 73a, 51-124 Wroclaw, Poland; sekretariat@wssk.wroc.pl

**Keywords:** diabetic foot syndrome, gene therapy, VEGF, HGF

## Abstract

Diabetic foot syndrome (DFS) is one of the most serious macroangiopathic complications of diabetes. The primary treatment option is revascularization, but complementary therapies are still being sought. The study group consisted of 18 patients diagnosed with ischemic ulcerative and necrotic lesions in DFS. Patients underwent revascularization procedures and, due to unsatisfactory healing of the lesions, were randomly allocated to two groups: a group in which bicistronic VEGF165/HGF plasmid was administered and a control group in which saline placebo was administered. Before gene therapy administration and after 7, 30, 90, and 180 days, color duplex ultrasonography (CDU) was performed, the ankle-brachial index (ABI) and transcutaneous oxygen pressure (TcPO2) were measured, and DFS changes were described and documented photographically. In the gene therapy group, four out of eight patients (50%) healed their DFS lesions before 12 weeks. During this time, the ABI increased by an average of 0.25 and TcPO2 by 30.4 mmHg. In the control group, healing of the lesions by week 12 occurred in six out of nine patients (66.67%), and the ABI increased by an average of 0.14 and TcPO2 by 27.1 mmHg. One major amputation occurred in each group. Gene therapy may be an attractive option for complementary treatment in DFS.

## 1. Introduction

Diabetes mellitus, a global epidemic, is currently diagnosed in half a billion people. It is estimated that this number will increase by one-quarter by 2030 and double by 2040 [1,2]. This makes the disease itself, as well as its complications, a challenge for every health care system in the world. Metabolically balanced diabetes requires a multidisciplinary approach from the medical side, as well as the involvement of the patient themselves. Things get complicated when one of these parts does not work. The result is long-term microvascular and macrovascular complications. The former is more common and includes neuropathy, nephropathy, and retinopathy, while macrovascular complications consist of cardiovascular disease, stroke, and peripheral artery disease (PAD) [3].

One of the most serious complications of diabetes is diabetic foot syndrome (DFS). By definition, it is an infection, ulceration, or tissue destruction of the foot in a person with currently or previously diagnosed diabetes, usually accompanied by neuropathy and/or PAD in the lower limb [4].

Neuropathy of the distal parts of the limbs leads to a reduction or loss of superficial sensitivity (pressure, touch), as well as a vibration sensation. The result is a reduction in pain sensation and a greater risk of injury and ultimately a hard-to-heal infection, resulting in a neuropathic diabetic foot [5,6].

In diabetic macroangiopathy, which can occur separately or coexist with neuropathy, there is multisegmental vascular involvement, especially below the knee, with impaired collateral circulation [7]. Ischemia in the form of PAD is associated with arteriosclerosis, which is much more common in those with diabetes and its course is more rapid. PAD is not only a significant risk factor for ischemic DFS, but also significantly worsens the prognosis of normal ulcer healing, in time and clinical effect [8].

The presence of DFS significantly Impairs quality of life, limiting life activities as well as psychosocial functioning. The prolonged healing process, often without favorable results, increases patient frustration and stress, and in many cases, leads to depression [9,10]. The extreme therapeutic failure is amputation of the infected limb. According to the literature, a lower limb amputation due to diabetes occurs every 20 s, and 85% of these cases are preceded by diabetic foot ulceration (DFU) [11,12]. People with diabetes have a 23 times higher risk of lower limb amputation than those without diabetes [13]. In addition, patients with this complication of diabetes have a significantly increased mortality rate compared to the healthy population [14,15,16]. Patients with DFU have a more than a twofold increase in mortality compared to patients with diabetes without ulceration, regardless of age, type and duration of diabetes, diabetes treatment, glycated hemoglobin level, history of lower limb amputation, and cumulative number of years of smoking [17].

On a societal level, the negative effects of the ever-increasing number of DFS and DFUs include the ever-increasing burden on national health care systems. In 2017, diabetes care costs in the US totaled USD 237 billion, with one-third of the costs attributable to the need to treat diabetic foot disease [18].

In accordance with the Hippocratic maxim that prevention is better than cure, special emphasis should be placed on primary prevention. It should include education on diabetes management, the principles of a healthy diet and appropriate physical activity, the prohibition of smoking, and direct foot protection against injury. A well-educated patient can not only have metabolically well-balanced diabetes, but also significantly delay or prevent the development of complications. Unfortunately, in many cases, when it is too late for primary prevention, diabetes and its complications, including DFS, must be intensively treated.

For ischemic lesions, patients are eligible for revascularization treatment. Vascular treatments, due to the peripherally dominant nature of the lesions, are mostly endovascular. They offer a chance of postoperative blood supply to the ischemic area and can accelerate the tissue healing process. However, these procedures are not always technically feasible or, when ultimately performed, do not have the intended beneficial effect. “Patients not eligible for revascularization” (NCR) are left with good diabetic metabolic control, use of statins, antiplatelet drugs, or smoking cessation. Such a conservative treatment strategy is not a part of active treatment. Therefore, in addition to improving endovascular techniques, new treatments for DFS are being sought for NCR patients.

Alternative treatments for the chronic limb-threatening ischemia (CLTI) patients are still being sought. Vasoactive drugs have not been proven to have a beneficial effect on wound healing and amputation rates in CLTI [19] and neither have epidural spinal cord stimulation (SCS) [20], fibrinolytic and defibrinogenating agents [21], or hyperbaric oxygen therapy (HBOT) [22]. Ozone therapy was also used, but due to the lack of effect, the additionally high doses of ozone, and the possible higher risk of cancer, this treatment regimen was withdrawn [23]. A study using transcutaneous muscle electrostimulation (TES) showed better blood flow in the anterior tibial artery in combination with prostacyclin than in the group treated with prostacyclin alone, but it did not prove a beneficial effect in patients with no-option CLTI based on study endpoints such as TcPO2 levels [24]. Dalla et al. used special wound dressings in patients with diabetes and after multiple minor amputations. The material stimulated extracellular matrix synthesis from fibroblasts to form a “neoderm” and avoided major amputation [25]. Lumbar sympathectomy (LSE), either surgical or chemical, can reduce rest pain in CLTI, but has no effect on the ABI, mortality, or amputation rates [26]. Research is currently underway on the effective use of “percutaneous deep leg vein arterialization” or the combination of open lower extremity deep vein bypass surgery with intraoperative destruction of venous valves. This treatment model eliminates rest pain and prevents obstruction [27].

For many years, attempts have been made to treat critical limb ischemia with gene therapy using plasmids encoding growth factors or autologous stem cells obtained from bone marrow or adipose tissue [28]. This therapy has been studied by the Department of Vascular Surgery of the Wroclaw Medical University since 2004 [29]. There are long-term studies of DFS gene therapy with a predominantly ischemic component in patients disqualified from revascularization treatment. To date, we have used growth factor plasmids administered intramuscularly with encouraging results [30]. We decided to use gene therapy as an adjunctive therapy for the treatment of DFS in a fairly homogeneous group of patients undergoing revascularization by open procedures using venous by-pass—the ‘in situ’ saphenous vein.

## 2. Materials and Methods

### 2.1. Patient Cohort

The group consisted of 18 patients (12 males and 6 females) with a mean age of 68 years (range 57–78), all suffering from DFS, and all patients had the following medical history: (a)A small ulceration, necrosis of the finger(s) or a non-healing wound after finger amputation.(b)All were categorized as grade IV on the Fontaine scale.(c)They had received previous intensive treatment with no improvement for at least 14 days.(d)They had a confirmed ischemic DFS variant with TcPo2 (transcutaneous oxygen pressure) ≤ 40 mmHg or ABI (ankle-brachial index) ≤ 0.4. Glycated hemoglobin (Hba1c) was determined in all patients at the start of treatment.(e)At 2–5 days after revascularization surgery, venous by-pass function was assessed using ultrasonography, and ABI and TcPo2 at ankle level were performed.

In 15 patients, the function of the graft was assessed well. In the remaining 3, despite the repair procedures, the venous by-pass became obstructed, and in 2 patients, a large amputation was required in the subsequent course, while in 1 (gene therapy group), a stationary ulceration persisted.

Patients were randomly assigned to a group in which a bicistronic plasmid containing growth factor genes, hepatocyte growth factor/ vascular endothelial growth factor (HGF/VEGF), was administrated or to a control group (saline administered intramuscularly). This was a double-blind trial in which only one of the study authors knew the formulation (he prepared it).

Observations were made after 7, 30, 90, and 180 days. Subsequently, the following were documented:(a)The status of the ischemic lesion (extent, demarcation, presence of necrosis, and infiltration of adjacent tissues/ulceration (extent, depth, activity-granulation, or anergy)). All lesions were documented with photographs and descriptions.(b)ABI—the ratio of the highest pressure from the anterior or posterior tibial artery to the highest systolic pressure in the arm.(c)TcPO2—the Medicap Precise 8001 device was used for these measurements.(d)CDU (color duplex ultrasonography).

### 2.2. Preparation and Administration of Plasmid DNA

The bicistron plasmid was developed as a result of our previous studies. Human cDNAs for HGF and VEGF165 were prepared as previously described [31]. Both cDNAs were cloned into the bicistronic pIRES plasmid using restriction enzymes. All pIRES/VEGF165/HGF plasmids thus obtained were purified and solubilized, and their pyrogenicity was excluded.

The plasmids were then injected intramuscularly into the ischemic limb above and below the knee region. Each patient in the gene-eligible group received 4 mg of bicistron plasmid (approximately 20 mL). Intramuscular injection sites were based on our previous studies and data from the literature [30,32]. The volume of each injection was about 0.25 mL (about 80 injections, 4 cm deep, into the ischemic muscles of the limb along each of the three main arterial trunks of the lower leg, and into the border zone between the angiosomes). The administration time did not exceed 20 min. Patients in the control group were administered saline instead of plasmid in the same manner.

### 2.3. Statistical Analysis

Statistica 13.3 (StatSoft, Krakow, Poland) was used for statistical analysis. The work presents the results classified as the so-called industry statistics, using both descriptive statistics (age, gender, yes/no data) and mathematical statistics.

Based on data from the literature, it was assumed that the sample error was about 2% and the confidence level was about 96%. The Shapiro–Wilk test showed that there was no basis for rejecting the hypothesis that the data did not follow a normal distribution, followed by Student’s *t*-test for comparisons between the obtained coefficients. Wilcoxon’s test was used for nonparametric analyses. The significance level was set at a *p*-value of less than 0.05.

## 3. Results

### 3.1. Group A (the Gene Therapy Group)

This group consisted of eight people, including five men and three women aged 59–78 years. Four people had necrosis of the toe, three people had ulceration of the foot or ankle (one case ending in major amputation), and one person had multiple necrotic lesions and ulcerations.

Prior to bicistron plasmid administration, four patients had an ‘in situ’ femoropopliteal venous graft, two patients had an ‘in situ’ femoropopliteal / sagittal venous graft, and the remaining two patients had an ‘in situ’ femoropopliteal venous by-pass with profundoplasty and/or iliac axis debridement.

Lesion healing in up to 12 weeks occurred in four patients (50% of the group), and above 12 weeks, there were no such cases.

The ABI increased from values in the range of 0.20–0.55 to values in the range of 0.5–0.9, with an average of 0.25.

TcPO2 increased from values in the range of 25–46 mmHg to values in the range of 58–91 mmHg, by an average of 30.4 mmHg.

The rest of the data on the amount of classical and endovascular procedures performed in the patients, the patency of the by-pass immediately after the procedures and after 3 months, and the presence of stationary ulceration are shown in Table 1. An example of the treatment effect of gene therapy is included in Figure 1

### 3.2. Group B (Control Group)

This group consisted of nine people, including six men and three women aged 59–78 years. Three people had necrosis of the toes, three patients had ulceration of the foot or lower leg, one person had multiple necrotic lesions and ulcerations, and two others had necrosis in the toe amputation ligation (one case ending in a major amputation).

Prior to saline administration, four patients had an ‘in situ’ femoropopliteal venous by-pass and five patients had an ‘in situ’ femoropopliteal venous by-pass with profundoplasty and/or iliac axis debridement.

Lesion healing in up to 12 weeks occurred in six patients (66.67% of the group), and above 12 weeks, there were no such cases.

The ABI increased from values in the range of 0.20–0.55 to values in the range of 0.45–0.85, by an average of 0.14, i.e., 0.11 less than in the gene therapy group.

TcPO2 increased from values in the range of 25–49 mmHg to values in the range of 52–85 mmHg, by an average of 27.1 mmHg, i.e., 3.3 mmHg less than in the gene therapy group.

The rest of the data on the amount of classical and endovascular procedures performed in the patients, the patency of the grafts immediately after the procedures and after 3 months, and the presence of stationary ulceration are shown in Table 2.

## 4. Discussion

DFS is one of the main practical problems encountered in angiology and vascular surgery. Owing to the huge prevalence of diabetes, its real epidemic, and the scale of its possible complications, it is one of the main causes of amputation and disability. The surgical approach, an attempt at revascularization, should always be considered. However, this is not always possible. The model of a multidisciplinary approach seems to be applicable and the most effective, with the care of a diabetologist, podiatrist, physiotherapist, and orthopedist, as well as an angiologist and vascular surgeon. Open surgery or endovascular revascularization is necessary in many cases. Their spectrum is very wide and depends on the specific situation. However, it should always aim to improve the supplementation of a limb or a specific angiosome, or even better perfused, with the hope that it will improve the neighboring angiosomes through collateral circulation. 

In addition to surgical revascularization, resection of necrotic tissues, drainage, load reduction, exercise, oxygen therapy, antibiotics, and metabolic therapy, it is necessary to search for new methods. This is particularly important for the group of patients in whom, due to the extent and advancement of the lesions, the possibilities of surgical treatment have been exhausted (NCR patients). Progress in vascular surgery, especially in endovascular interventions, does not allow the treatment of all patients with lower limb ischemia in the course of diabetes, especially since the pathologies usually concern small vessels, often more strongly expressed in some angiosomes, leaving others in a better condition. If surgical revascularization is not possible, conservative therapy remains, such as pharmacotherapy, load reduction, and improvement of their general condition, which is often insufficient, although necessary.

Gene therapy that stimulates the formation of new, even small, vessels seems to be an interesting concept. Stimulation of the formation of small vessels and collaterals allows the development of collateral circulation in the long axis of the limb as well as between angiosomes within the lower leg and foot. Effectively improving the influx to the impaired angiosomes. Gene therapy is associated with a few mild side effects. There is a limited pool of complications, and the condition of the limb does not deteriorate. This is confirmed by our previous experiences [33]. 

In addition to VEGF and HGF, other growth factors have been tried in gene therapy. Creager et al. used hypoxia-inducible factor 1-alpha (HIF-1α)-encoding for treatment, but this did not prove to be an effective treatment for patients with intermittent claudication [34]. The same factor was studied by Rajagopalan et al., who described its good tolerability in patients with CLTI but noted that its use requires further study [35]. Grossman et al. evaluated a treatment encoding for the developmental endothelial locus-1 (Del-1) protein and showed no significant differences of such treatment with the control group [36]. In another study, Hammad et al. used a treatment that encoded for stromal cell-derived factor-1 (SDF-1), and such a combination with revascularization did not improve the CLTI patient’s condition [37].

The process of neovascularization requires the cooperation of many cells and proteins. Based on the literature as well as our own experience, we chose HGF and VEGF for gene therapy, which we believe is the most optimal combination [38,39]. VEGF is a key initiator of the whole process by stimulating endothelial cell proliferation and migration. VEGF-A in excessive amounts has been shown to initiate new vessel formation in vivo in damaged or ischemic tissue [40,41]. Muona et al. conducted a study using the VEGF plasmid on a group of 54 patients. They achieved an impressive follow-up period of 10 years. They proved that administration of the gene did not cause any threat in the form of an increased risk of cancer, diabetes, or other diseases [42]. In 2003, Shyu et al. proved that VEGF gene therapy is not only safe but also effective. In the group of patients receiving the gene, the ABI increased significantly, and angio-MR studies showed an improvement in the vascularity of the limbs [43,44].

The whole process, its stability, and its lasting effect require other factors, such as HGF. HGF has been shown to be a potent activator of cell proliferation, and in addition, compared to other growth factors, it does not induce inflammation, which is its unique value and may affect the efficiency of the angiogenesis process [38,39].

Considering these successes, we decided to use the synergistic action of the two proangiogenic factors. In order to optimize gene therapy as much as possible, vascular mapping and ultrasound were performed prior to VEGF/HGF administration. The choice of the intramuscular route of administration was based on its greatest efficacy in promoting angiogenesis from the available literature data and our own experience [45,46]. The first injections were made 2 cm above the site of obstruction, and the subsequent injections were regularly spaced along the blood vessel at least 2 cm apart.

In this study, in the VEGF/HGF gene treatment group, lesion healing by week 12 occurred in four patients (50% of the group), the ABI increased from values in the range of 0.20–0.55 to values in the range of 0.5–0.9, by an average of 0.25, and TcPO2 increased from values in the range of 25–46 mmHg to values in the range of 58–91 mmHg, by an average of 30.4 mmHg. In patients in the control group, lesion healing by week 12 occurred in six patients (66.67% of the group), the ABI increased from values in the range of 0.20–0.55 to values in the range of 0.45–0.85, by an average of 0.14, 0.11 less than in the gene therapy group. TcPO2 increased from values in the range of 25–49 mmHg to values in the range of 52–85 mmHg, an average of 27.1 mmHg, 3.3 mmHg less than in the gene therapy group. In our other study, bicistronic VEGF/HGF gene therapy achieved satisfactory results in the treatment of DFS. After 90 days of therapy, significant reductions in pain and increases in the ABI were observed [33]. As expected, the inclusion of this therapy in patients undergoing peripheral artery bypass graft surgery improved the outcomes. It can be said that the combination of conventional therapy (by-passes) and bigenotherapy significantly improved the effect of such comprehensive treatment. A limitation of the above paper may be the number of participants in the study and thus the inability to carry out a more accurate statistical analysis, including calculation of the correlation coefficient between some characteristics and treatment outcomes.

In addition to gene therapy, other mechanisms to enhance neovascularization should be considered. It has become apparent in recent years that there are extensive similarities between the development of nerves and blood vessels. Both are branched structures that require guidance to reach their proper targets. A variety of molecules that were previously thought to be restricted to axonal guidance processes have now been shown to modulate blood vessel guidance as well. These insights are of significance as they could have potential therapeutic applications [47,48]. The fact that molecules such as slits, semaphorins, and netrins may modulate physiological, as well as pathological, angiogenesis is likely to lead to the development of novel strategies to promote or inhibit angiogenesis [49,50]. It should be emphasized, however, that there are no published works on the use of these molecules in therapeutic neoangiogenesis in experimental and clinical conditions in the human population. Clinical application requires further experimental studies in vitro and in vivo [51].

In our study, treatment outcomes after complementary gene therapy were comparable to the control group. The small size of the study group does not allow us to claim that the changes are statistically significant. This is related to the assumption of obtaining as homogeneous a group as possible of patients with diabetes, burdened by DFS, critical ischemia, and after revascularization procedures. The results are certainly not worse in the study group (after gene therapy) than in the control group and no adverse events were observed. A precise determination of the synergistic effect of the therapy will be possible on a larger statistically significant group. With the safety of this method proven, the study can be continued.

## 5. Conclusions

A primary prerequisite for the healing of ulcerated tissues is to ensure that they have a proper blood supply. Revascularization treatments are an absolutely essential, primary intervention in DFS with significant macroangiopathy and limb ischemia. When gene therapy was additionally applied immediately after revascularization treatment, there was certainly no worsening of the treatment outcome, but rather better results were achieved with a reduction in the healing time. DFS is a severe, poor prognosis disease that is difficult to treat. A multifaceted approach is a prerequisite for successful treatment. It is necessary to try to broaden the range of therapeutic options and not to give up on any method.

## Figures and Tables

**Figure 1 genes-14-01968-f001:**
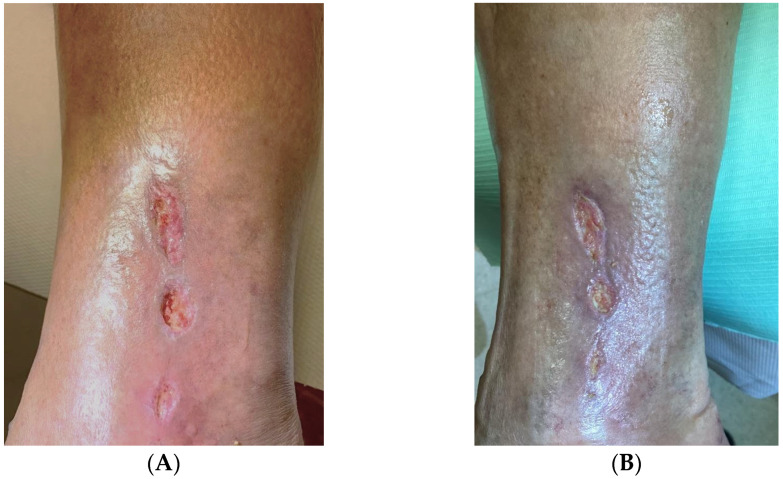
Ankle ulceration in a Group A (the gene therapy group) (**A**) before treatment and (**B**) after 12 weeks.

**Table 1 genes-14-01968-t001:** The gene therapy group.

Sex	W	M	M	M	M	M	W	M	W
Age	66	60	78	57	77	75	59	72	63
Hba1c	9.0	8.8	9.6	10.3	8.5	9.9	10.4	13.3	11.2
Type of trophic changes	1	1	1	2	2	2	3	4	4
ABI	0.33	0.55	0.3	0.5	0.4	0.31	0.3	0.2	0.35
ABI after 3 months	0.65	0.9	0.67	0.72	0.7	0.6	0.72	-	0.5
TcPO2 [mmHg]	35	30	29	40	33	32	46	25	41
TcPO2 after 3 months [mmHg]	76	89	68	80	48	91	75	-	58
Healing up to 12 weeks	Y	Y	Y	Y	N	Y	Y	N	N
Healing more than 12 weeks	N	N	N	N	N	N	N	N	N
Type of treatment	1	1	3	1	3	3	1	3	3
Patency of a venous graft after surgery	Y	Y	Y	Y	Y	Y	Y	N	Y
Patency of a venous graft after 3 months	Y	Y	Y	Y	N	Y	Y	N	Y
Stationary ulceration	N	N	N	N	Y	N	N	N	Y
Major amputation	N	N	N	N	N	N	N	Y	N
Amount of previous open surgical revascularizations	1	0	0	0	2	0	1	0	0
Amount of previous endovascular revascularizations	3	1	2	2	1	3	4	1	2

Type of trophic changes: 1—necrosis of the toe(s); 2—ulceration of the foot/ankle; 3—several lesions; 4—necrosis after amputation of the toe. Type of treatment: 1—“in situ” femoropopliteal venous by-pass; 2—“in situ” femoro-crural (tibial or peroneal) distal venous by-pass; 3—“in situ” femoropopliteal venous by-pass with profundoplasty and/or iliac artery endarterectomy.

**Table 2 genes-14-01968-t002:** Control group.

Sex	W	M	M	W	M	W	M	M
Age	60	65	61	72	70	78	62	59
Hba1c	8.4	9.7	8.8	9.2	9.4	10.1	12.3	10.6
Type of trophic changes	1	1	1	1	2	2	2	3
ABI	0.45	0.55	0.2	0.4	0.42	0.43	0.35	0.45
ABI after 3 months	0.59	0.7	0.45	0.66	0.85	-	0.5	0.64
TcPO2 [mmHg]	34	30	26	49	25	29	40	35
TcPO2 after 3 months [mmHg]	70	65	60	71	82	-	85	52
Healing up to 12 weeks	Y	Y	N	N	Y	N	N	Y
Healing more than 12 weeks	N	N	N	N	N	N	N	N
Type of treatment	1	3	1	2	3	2	1	1
Patency of a venous graft after surgery	Y	Y	Y	Y	Y	N	Y	Y
Patency of a venous graft after 3 months	Y	Y	Y	Y	Y	N	Y	Y
Stationary ulceration	N	N	Y	Y	N	N	Y	N
Major amputation	N	N	N	N	N	Y	N	N
Amount of previous open surgical revascularizations	**0**	**0**	**0**	**1**	**1**	**0**	**1**	**0**
Amount of previous endovascular revascularizations	**2**	**3**	**2**	**1**	**4**	**5**	**4**	**3**

Type of trophic changes: 1—necrosis of the toe(s); 2—ulceration of the foot/ankle; 3—several lesions; 4—necrosis after amputation of the toe. Type of treatment: 1—“in situ” femoropopliteal venous by-pass; 2—“in situ” femoro-crural (tibial or peroneal) distal venous by-pass; 3—“in situ” femoropopliteal venous by-pass with profundoplasty and/or iliac artery endarterectomy.

## Data Availability

The data are available upon request.

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
