# Peer review of "Complementary Gene Therapy after Revascularization with the Saphenous Vein in Diabetic Foot Syndrome"

_genes, 2023, doi:10.3390/genes14101968_

Round 1

Reviewer 1 Report

The article “Complementary gene therapy after revascularization with the 2 saphenous vein in diabetic foot syndrome” was submitted for review. Metabolically balanced diabetes requires a multidisciplinary approach. One of the most serious complications of diabetes is diabetic foot syndrome (DFS). By definition, it is an infection, ulceration, or destruction of the tissues of the foot in a person with currently or previously diagnosed diabetes, usually accompanied by neuropathy and/or in the lower extremity. In the elderly, a lower limb amputation due to diabetes occurs every 20 seconds, and 85% of these are preceded by diabetic foot ulceration (DFU). In addition to improving endovascular techniques, new treatments are being sought for patients with DS. For many years, attempts have been made to treat critical limb ischemia with therapies using plasmids encoding growth factors or autologous stem cells derived from bone marrow or adipose tissue. The authors decided to use gene therapy as an adjunctive therapy for the treatment of DFS in a fairly homogeneous group of patients who underwent revascularization with open procedures using venous bypass - saphenous vein "in situ".

The group consisted of 8 people, including 5 men and 3 women aged 59-78 years. 4 people had necrosis of the toe, 3 people had ulceration of the foot or ankle (1 case ending in major amputation), 1 person had multiple necrotic lesions and ulcerations. The group consisted of 9 people, including 6 men and 3 women aged 59-78 years. 3 people had necrosis of the toes, 3 patients had ulceration of the foot or lower leg, 1 person had multiple necrotic lesions and ulcerations, and 2 others had necrosis in the toe amputation ligation (1 case ending in a major amputation).

Adequate and modern methods of statistical data processing were used. It was concluded that the primary condition for the healing of ulcerated tissue is to ensure an adequate blood supply. Revascularization is an absolutely necessary primary intervention for DFS with significant macroangiopathy and limb ischemia. When gene therapy was additionally applied immediately after revascularization treatment, of course, there was no worsening of the treatment outcome, but better results were achieved with a reduction in healing time. The article is illustrated with 1 figure and 2 tables. The bibliography consists of 24 sources. A very relevant and timely publication dedicated to the important problem of diabetic foot.

Conclusion. It is advisable to add information to the discussion section and supplement the list of references with modern sources.

Author Response

Thank you very much for reviewing our work. We have taken all comments into account and edited our manuscript accordingly. We have added a paragraph in the introduction on treatment alternatives in DFS. Major changes have also occurred in the discussion.

Reviewer 2 Report

The article genes-2637417 entitled ‘Complementary gene therapy after revascularization with the 2 saphenous vein in diabetic foot syndrome’ by Barć et al. describes effect of VEGF165/HGF plasmid genetic therapy (8 patients) vs. saline placebo injections (9 patients) on FDS lesions, ankle-brachial index (ABI), transcutaneous oxygen pressure 32 (TcPO2) after 12 weeks of observation in 18 patients diagnosed with ischemic ulcerative and necrotic lesions in FDS.  Authors found a modest effect of gene therapy with 50% healing of FDS lesions vs 66% control and a not significant increase in ABO and TcPO2 in gene therapy group compared to placebo. In treatment and placebo group one patient has an amputation. The major disconnect in this study is the claim ‘Gene therapy may be an attractive option for complementary treatment in FDS’ (Abstract), while this gene therapy composition did not improved patients’ conditions. Undoubtedly, the progress in FDS treatment is in developing of gene therapy; however, this progress could be achieved by detailed analysis of why VEGF165/HGF plasmid composition was not successful.

Introduction is lengthy and is appropriate for review. Instead of lengthy discussions of economic burden of diabetes and FDS, authors should describe what are current state of art in gene therapy of revascularization disease, efficacies of various  genes used for these therapies and why VEGF165/HGF is a promising candidate target.  IT is also worthy to discuss the influence of delivery systems and regimen of treatments and how these factors may influence efficacy of gene therapies. Many abbreviations were not de-abbreviated (FDS, VEGF165/HGF).

Innervation precede vascularization (e.g.PMID: 23475066) , therefore, the use of axon guiding molecules in genetic therapies should be discussed.

 Statistic analysis should be preformed between treatment and control groups. Where any of the treatment parameters correlated with age? HbA1C levels? These correlation analyses should be included. In fact, there are is a very limited information about patients. VEGF can exhibit sex different wound healing effects, even though the number of patients was very limited this issue need to be at least discussed.

 Discussion is not discussing results. The limitation of this study in terms of therapeutic efficacy was not discussed in context of gene composition, methods of delivery, quality of vectors, timing to the treatment, etc.

English is fine, but many terms are abbreviated without explanations.

 Context of  the Introduction and Discussion needs major improvements.

Author Response

Thank you very much for reviewing our work. We have taken all comments into account and edited our manuscript accordingly. We have added a paragraph in the introduction regarding alternative treatments for DFS. There have also been major changes to the discussion, which we are sending in the attached

Round 2

Reviewer 2 Report

The authors did not address the major disconnect: their conclusions are not supported by the data. They themselves acknowledge that the data lack statistical significance. While they may suggest, based on the facts presented, that this study provides a rationale for further research in a larger group of patients, it's worth noting that, as of now, there is no discernible difference between the control and gene therapy groups.

'In our study, treatment outcomes after complementary gene therapy were better than 336 in the control group.'

Figure 1 attempts to illustrate the benefits of gene therapy, but it displays pictures of different sizes, making it challenging to accurately gauge wound size. Variations in lighting conditions also hinder the ability to determine whether the images represent the same patient. Additionally, the figure from the control group is conspicuously absent.

It is understandable.

Author Response

In our study, treatment outcomes after complementary gene therapy were comparable to the control group. The small size of the study group does not allow us to claim that the changes are statistically significant. This is related to the assumption of obtaining as homogeneous a group as possible of patients with diabetes, burdened by DFS, critical ischaemia and after revascularisation procedures. The results are certainly not worse in the study group (after gene therapy) than in the control group and no adverse events were observed. A precise determination of the synergistic effect of the therapy will be possible on a larger statistically significant group. With the safety of this method proven, the study can be continued.

Figure 1 has been corrected. The addition of graphics is for illustrative purposes and not for comparison with the control group